# Epilepsy Surgery for Skull-Base Temporal Lobe Encephaloceles: Should We Spare the Hippocampus from Resection?

**DOI:** 10.3390/brainsci8030042

**Published:** 2018-03-12

**Authors:** Firas Bannout, Sheri Harder, Michael Lee, Alexander Zouros, Ravi Raghavan, Travis Fogel, Kenneth De Los Reyes, Travis Losey

**Affiliations:** 1Department of Neurology, Loma Linda University Health, Loma Linda, CA 92354, USA; TLosey@llu.edu; 2Department of Radiology (Division of Neuroradiology); Loma Linda University Health, Loma Linda, CA 92354, USA; SHarder@llu.edu; 3Department of Radiology, Los Angeles County and University of Southern California Medical Center, Los Angeles, CA 90033, USA; mjl_202@med.usc.edu; 4Department of Neurosurgery, Loma Linda University Health, Loma Linda, CA 92354, USA; AZouros@llu.edu (A.Z.); KDelosreyes@llu.edu (K.D.L.R.); 5Department of Pathology, Human Anatomy & Neurosurgery, Loma Linda University Health, Loma Linda, CA 92354, USA; RRaghavan@llu.edu; 6Loma Linda Physical Medicine and Rehabilitation, Neuropsychology; Loma Linda University Health, Loma Linda, CA 92354, USA; TFogel@llu.edu

**Keywords:** temporal lobe epilepsy, encephalocele, meningoencephalocele, tailored surgery

## Abstract

The neurosurgical treatment of skull base temporal encephalocele for patients with epilepsy is variable. We describe two adult cases of temporal lobe epilepsy (TLE) with spheno-temporal encephalocele, currently seizure-free for more than two years after anterior temporal lobectomy (ATL) and lesionectomy sparing the hippocampus without long-term intracranial electroencephalogram (EEG) monitoring. Encephaloceles were detected by magnetic resonance imaging (MRI) and confirmed by maxillofacial head computed tomography (CT) scans. Seizures were captured by scalp video-EEG recording. One case underwent intraoperative electrocorticography (ECoG) with pathology demonstrating neuronal heterotopia. We propose that in some patients with skull base temporal encephaloceles, minimal surgical resection of herniated and adjacent temporal cortex (lesionectomy) is sufficient to render seizure freedom. In future cases, where an associated malformation of cortical development is suspected, newer techniques such as minimally invasive EEG monitoring with stereotactic-depth EEG electrodes should be considered to tailor the surrounding margins of the resected epileptogenic zone.

## 1. Introduction

An encephalocele refers to the protrusion of brain tissue, meninges, and cerebral spinal fluid (CSF) through a calvarial or skull base defect. Most of these lesions are due to congenital defects, although they can be acquired following trauma, surgery, or infection [1,2]. When associated with temporal lobe seizures, encephaloceles are associated with bony defects in the middle cranial fossa [3,4]. Associated defects have also been localized to the petrosal bone [5,6], cribriform plate [7], and diffusely throughout the skull. 

Imaging is often reported to be normal, with the skull-base lesion noted only upon reassessment of initial neuroimaging [3,5,6,8,9]. Magnetic resonance imaging (MRI) and high-resolution CT scan imaging of the skull base allow for identification of these rare lesions and provide critical information needed to decide if a surgical approach is required as part of the treatment plan. High-resolution CT imaging has been shown to detect bony defects associated with temporal lobe encephaloceles that are not evident on MRI [3,8,10].

Patients with skull-base temporal encephaloceles may rarely present with seizures [3,4]. These seizures are usually due to involvement of the neocortex, and are frequently refractory to medical therapy [4,10,11]. Suggested mechanisms include tissue traction, gliosis, or associated dysplasia [7]. Associated cranial abnormalities are reported in only 15% of cases. These include band heterotopia, nodular heterotopia, diffuse cortical dysplasia, and schizencephaly [7]. 

Resection of the encephalocele with temporal neocortical excision (lesionectomy) or performing an anterior temporal lobectomy with amygdalohippocampectomy (ATLAH) has been reported to be effective in controlling seizures [3,4,7,8,10,12,13,14]. For example, in 18 cases collected by Faulkner et al., 2010, 56% underwent local excision and 44% underwent a wider excision with a lobectomy. The majority of extra temporal cases (33%) are treated with local excision, while the majority of temporal lobe cases (67%) were treated with lobectomy. There was no difference in the reported seizure freedom rates postoperatively (100% seizure freedom in all patients).

We reviewed 44 patients with epilepsy and surgically treated temporal encephalocele as described in the English-language medical literature (Table 1). Only one case report of bitemporal encephalocele was found, and the remaining cases were evenly divided between the left (54%) and the right (52%) temporal side. Sixty-three percent of cases were treated conservatively with lesionectomy or anterior temporal lobectomy (unclear extension of resected tissue), and 34% of cases were treated with ATLAH. The largest case report series were compiled by Panov et al., 2016 (six patients) [14] and Saavalainen et al., 2015 (twelve patients) [15].

Surgical approaches vary depending on seizure type, origin of encephalocele (congenital vs. acquired), location (temporal vs. extra-temporal), the type of brain lesion seen on MRI (with associated signal abnormality vs. none), and concordance of studies (scalp EEG MRI, positron emission tomography (PET) and neuropsychological testing). Most recently, stereotactic-depth EEG electrode implantation was used in one case report of temporal lobectomy and amygdalectomy alone, sparing the hippocampus, with reported seizure freedom [16].

## 2. Case Series

We report two cases of temporal lobe encephalocele, one congenital and another traumatic, which were treated with local excision and reparation of the encephalocele. Both cases underwent temporal lobe neocortical resection, sparing the hippocampus. Both patients have remained with Engel class-I surgical outcome (42 months for case 1 and 23 months for case 2). 

### 2.1. Case 1: 24-Year-Old Ambidextrous, Bilingual (English and German) Man with Seizures Since 22 Years of Age. Seizure Semiology: Ictal Expressive Aphasia, Right Forced Head and Eyes Deviation, Lip-Smacking, and Sensation of Fear

Preoperative MRI (Figure 1A,B) and maxillofacial CT scan (Figure 1C) showed a large temporo-sphenoidal encephalocele with a defect in the floor of the middle cranial fossa with herniation of gliotic temporal and frontal opercular tissue into the defect.

Scalp video-EEG monitoring captured five complex partial seizures, localized broadly to the left fronto-temporal channels (T3/T5/O1 and FP1/F3/F7). The ictal pattern was from EEG electrodecremental response, followed by 1.5–2 Hz rhythmic activity then 12–15 Hz low amplitude rhythmic activity (maximally at T5/O1), which is not typical for medial temporal lobe onset seizure but rather for neocortical onset seizure [22] (Figure 2). Electrocardiogram (ECG) correlated with tachycardia up to 120 bpm. 

Intraoperative ECoG monitoring with three 2 × 4 grids were used to cover the left frontal operculum, anterior temporal tip containing the dysplastic tissue, and mid-posterior/lateral temporal cortex. One 1 × 8 strip was placed in the inferior medial and anterior temporal lobe. The strip and grid located in the inferior and anterior temporal lobes revealed occasional dysmorphic spikes anteriorly. 

Epileptiform discharges were anteriorly located with the temporal pole and possibly the frontal opercular region. There was no evidence of epileptiform discharges coming from the medial-posterior temporal lobe strips. 

Surgical approach included the insertion of a lumbar cerebrospinal fluid intrathecal drain, harvest of abdominal fat through a separate incision, left temporal craniotomy and temporal lobectomy with intraoperative electrocorticography, fronto-zygomatic craniotomy, and repair of left sphenonasal encephalocele with microscopic dissection. The ECoG findings allowed for identification of a 3-cm margin from the temporal tip for partial resection. 

Surgical pathology of the resected tissue showed benign cerebral parenchyma and focal neuronal heterotropia within the deeper portion of the white matter (Figure 3). There was no evidence of gliosis or inflammation. There were no abnormal inclusions, myelin-related or metabolic abnormalities, or neoplastic changes. 

Outcome: Seizure frequency prior to surgery was 4–5 times per week. He has remained seizure-free for 42 months since his surgery, during which levetiracetam was discontinued and lacosamide was reduced from 200 mg bid to 100 mg bid. Patient was able to continue his collegiate studies and scored 90th percentile on his *Medical College Admission Test* (MCAT). Post-operative MRI showed preserved left hippocampus (Figure 4A) and the left temporal surgical bed (Figure 4B). The amygdala was resected as well (Figure 4C). 

### 2.2. Case 2: 66-Year-Old Right-Handed Man with Intractable Epilepsy since the Age of 43 Years, after He Was Struck on Forehead at Work by a Pump While Pumping Oil from a Compressor. Seizure Semiology Consisted of Dizziness, Starring Off, Lip-Smacking, Excessive Drooling, Right Hand Rolling Movements, and Occasional Bilateral Tonic-Clonic Seizures

Preoperative MRIs in 2009 and 2013 were reported as normal (1.5 Tesla). Preoperative high-resolution 3.0 Tesla MRI showed herniated right temporal pole encephalocele (Figure 5). Both hippocampi appeared normal and symmetric on MRI (Figure 5C). 

Scalp video-EEG monitoring captured nine stereotyped electroclinical seizures (starring off, lip-smacking with occasional right manual automatisms). Ictal onset originated from the right anterior and mid-temporal channels. The electrographic ictal pattern consisted of rhythmic 2–3 Hz activity, suggesting neocortical rather than mesial temporal onset seizure pattern [22] (Figure 6). 

Neuropsychological testing showed relatively intact neuropsychological functioning with some areas of relative weaknesses, including evidence of weaker nonverbal (visual) new learning and memory compared to his auditory (verbal) new learning and memory as well as significant visuoconstructional weaknesses. The absence of any obvious nonverbal (visual) memory impairments provided support for the absence of any right mesial temporal dysfunction and the greater likelihood of seizure control with selective resection of the inferior temporal lobe.

Surgical approach included right temporal craniotomy, repair of lateral sphenoid encephalocele with repair of dural defect and rotational temporalis muscle and fascial flap, partial resection of right middle and inferior temporal gyrus for resection of seizure focus, and lumbar drain placement.

To address the seizure focus, the middle and inferior temporal gyri were identified and cauterized. Then, a corticotomy was performed followed by partial resection of the middle and inferior temporal gyri. 

Outcome: Seizure occurred several times per month before surgery. He has remained seizure-free for 23 months since surgery. Seizure medications were reduced, during which lamotrigine was reduced from 900 mg to 600 mg daily and pregabalin was switched to gabapentin (from insurance perspective). Patient also has early onset Parkinson’s disease and developed insomnia and restless leg syndrome. Clonazepam was added at 1 mg at night. Pathology was not done due to the use of cauterized resection. A post-operative MRI (Figure 7) showed the surgical cavity in the region of the previously noted encephalocele with a small residual fluid collection in the right infra-temporal fossa (Figure 7B).

## 3. Discussion

Meningoencephaloceles can be associated with microscopic congenital malformations. As such, it is important to realize that the meningoencephalocele may be the “the tip of the iceberg” and the diagnosis should lead one to carefully search for evidence of an underlying malformation of cortical development. However, seizure rates and associated cortical congenital abnormalities are probably more uncommon in this rare form of encephaloceles compared to non-skull base encephaloceles (cranial-vault) seen in pediatric patients, which commonly have multifocal cortical abnormalities and generalized epilepsy [23]. 

The optimal surgical strategy for intractable epilepsy patients with a temporal encephalocele is not well established. Tailored surgery of temporal pole encephaloceles was described by Giulioni et al. [10]. As it has been reported before, local excision (in extra-temporal cases) has a similar chance of seizure freedom as lobectomy (in temporal cases), regardless of the location of the encephalocele (Faulkner). In our review of nearly 44 cases published so far, a conservative surgical approach was seen more often than the larger resection of the anterior temporal lobe, including the amygdala and the hippocampus combined.

We describe here two cases of temporal lobe encephaloceles and related epilepsy in which tailored temporal lobe resection was performed. The left (dominant) hippocampus was preserved in case 1 due to the demanding high cognitive/memory function of a college student, and the right (non-dominant) hippocampus was preserved in case 2 due to patient choice and concern of worsening memory in the setting of his advanced age. 

In the presence of electrographic or radiographic evidence of wide mesial structure involvement, the approach of ATLAH is reasonable, as shown by Panov et al., who reported four out of six surgical cases of temporal encephaloceles treated with ATLAH. However, two of those six cases were treated, similarly to our cases, with lesionectomy and temporal lobe disconnection. Our cases also illustrate that temporal lobe encephaloceles can be treated successfully with a conservative surgical approach sparing the mesial temporal structures. 

The result of post-operative seizure freedom after encephalocele repair with only limited temporal lobe resection, sparing the hippocampus, demonstrates the possibility and the importance of preserving the vital memory function of the hippocampus without jeopardizing post-surgical seizure freedom. 

As shown by De Souza et al., most recent advances in stereotactic depth electrode implantation may provide us with better options for intracranial EEG monitoring in future cases, which would allow us to identify the epileptogenic zone more precisely for better outcome 

## Figures and Tables

**Figure 1 brainsci-08-00042-f001:**
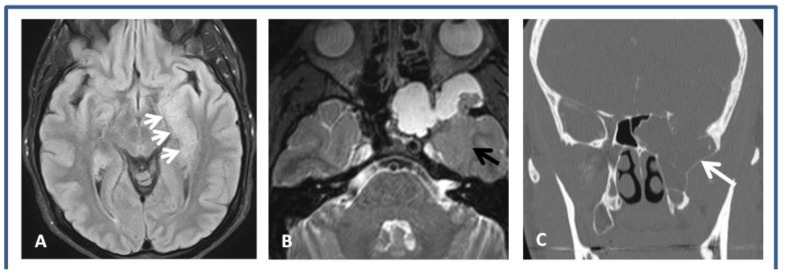
Preoperative brain imaging. (**A**) Axial FLAIR sequence showing increased signal in the left medial temporal lobe (small white arrows). (**B**) Axial T2 sequence shows herniated intracranial contents extending through the skull base defect. There is thickening of the regional cortex (black arrow). (**C**) Coronal CT maxillofacial showing the large left temporo-sphenoidal encephalocele (white arrows).

**Figure 2 brainsci-08-00042-f002:**
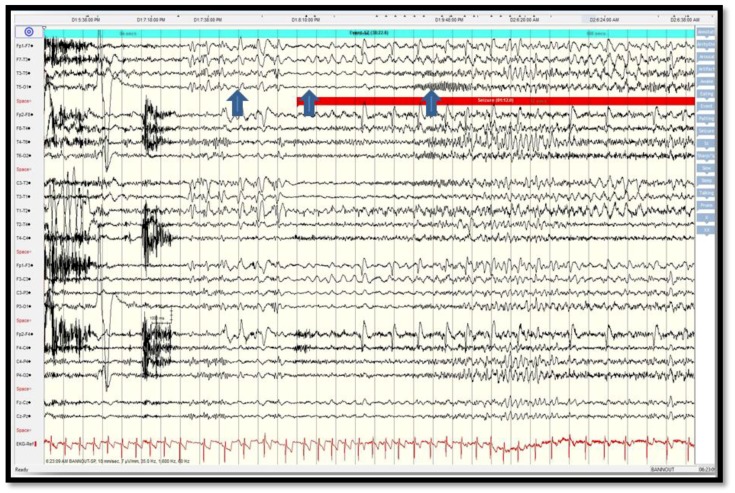
Scalp EEG recording showing broad periodic left fronto-temporal sharp waves (arrow 1 from the left) followed by attenuation at T3/T5 and T3/T1 (arrow 2) then low amplitude rhythmic beta frequencies at T5/O1 (arrow 3).

**Figure 3 brainsci-08-00042-f003:**
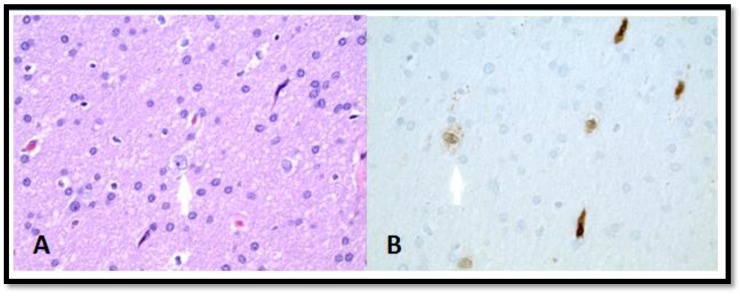
Histology of the resected left temporal lobe tissues. (**A**) Heterotopic white matter neurons HE stain × 400. (**B**) Heterotopic white matter neurons (5 in the field) highlighted by NeuN immunostain × 400.

**Figure 4 brainsci-08-00042-f004:**
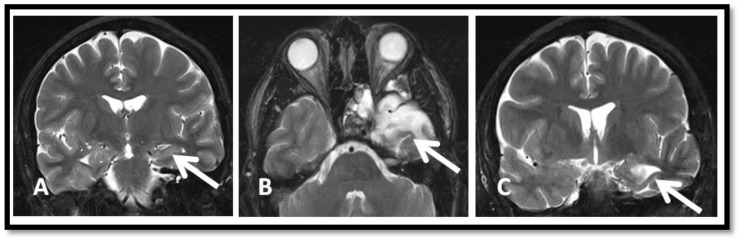
One year post operative brain MRI. (**A**) Coronal T2 sequence showing preserved left hippocampus (arrow). (**B**) Axial T2 sequence showing resected left temporal pole (arrow). (**C**) Coronal T2 sequence showing left amydalectomy (arrow).

**Figure 5 brainsci-08-00042-f005:**
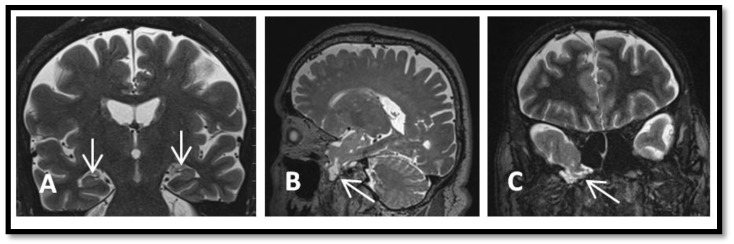
Preoperative brain MRI. (**A**) Coronal T2 sequence showing symmetrical bilateral hippocampi (vertical arrows). Coronal and Sagittal T2 sequence (**B**,**C**) showing small right sphenotemporal meningocele with small encephalocele and focal encephalomalacia of the inferior right temporal cortex (arrows).

**Figure 6 brainsci-08-00042-f006:**
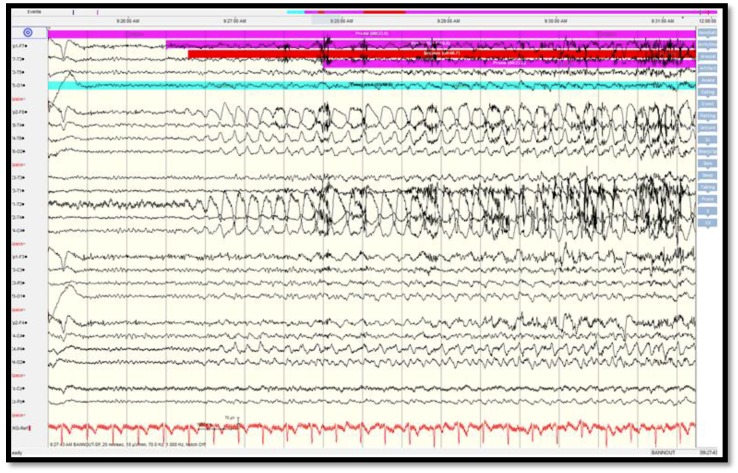
Scalp EEG recording showing right anterior temporal evolving 2–3 Hz ictal rhythm (arrows).

**Figure 7 brainsci-08-00042-f007:**
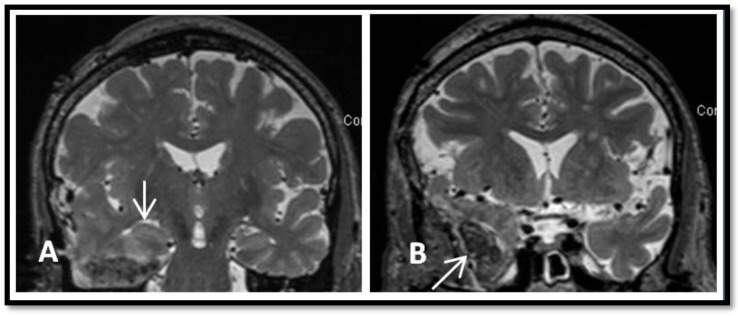
A post-operative brain MRI showing postsurgical changes related to resection of an encephalocele and the floor of the right middle cranial fossa. (**A**) Preserved right hippocampus (vertical arrow). (**B**) A surgical defect at the anterior aspect of the right temporal lobe (arrow).

**Table 1 brainsci-08-00042-t001:** Reported surgically treated temporal lobe encephaloceles in patients with epilepsy.

Authors, Year	Age, Sex	Location	Congenital or Acquired	Surgical Approach	Pathology	Seizure Outcome/Duration
Ruiz Garcia et al., 1971 [9]	30, F	L temporal	Congenital	ATL	Gliosis and fibrosis	Free/NA
Hyson et al., 1984 [5]	40, W	R temporal	Acquired (after right mastoidectomy)	Lesionectomy	NA	Free/NA
	12, M	R temporal	Acquired (after trauma)	ATL	Mixed astrocytoma grade I and oligodendroglioma.	NA
	37, F	L temporal	Congenital	ATL	Gliosis	Free/3 y
Rosenbaum et al., 1985 [17]	38, F	R temporal	Congenital	ATL + AH	NA	Only Aura
	30, F	R temporal	Congenital	ATL + AH	NA	Free/NA
Whiting et al., 1990 [18]	18, F	R temporal	Congenital	ATL	Meningo-angiomatosis.	Only Aura
Le Blanc et al., 1991 [13]	37, F	L temporal	Congenital	ATL + AH	Gliosis	Free/NA
	36, M	L temporal	Congenital	ATL + AH	Gliosis	Free/NA
	26, M	L temporal	Acquired	ATL + AH	Gliosis	Free/NA
Wilkins et al., 1993 [19]	36, F	R temporal	Congenital	ATL	Gliosis	Free/18 Mo
Mulcahy et al., 1997 [20]	25, F	L temporal	Congenital	Lesionectomy	NA	NA
Yang et al., 2004 [6]	46, M	Bitemporal	Congenital	Lesionecotmy	Inflamed neuroglial	Free/NA
Byrne et al., 2010 [21]	26, M	L temporal	Congenital	Lesionectomy	Astrocystotis	Free/7 y
	42, F	L temporal	Congenital	ATL + AH	Astrocystotis	Free/2 y
	57, M	R temporal	Congenital	ATL + AH	Astrocystotis	Free/1 y
Aquilina et al., 2010 [12]	14, F	L temporal	Congenital (14)	ATL + AH	Diffuse temporal gliosis involving the HC + microdysgenesis of the amygdala.	Free
Abou-Hamden et al., 2010 [3]	39, F	L temporal	Congenital	ATL	Gliosis	Free/22 Mo
	26, F	L temporal	Congenital	ATL	Gliosis	Free/12 Mo
	26, F	L temporal	Acquired (forceps delivery)	ATL	Gliosis	Free/12 Mo
Giulioni et al., 2014 [10]	41, M	L temporal	Congenital	ATL	Microdysgenesis	Free/5 y
	63, M	L temporal	Congenital	ATL	Microdysgenesis	Free/4 y
Gasparinin et al., 2014 [11]	20, M	L temporal	Congenital	Lesionectomy	Mild gliosis	Free/20 Mo
Shimada et al., 2015 [4]	21, M	L temporal	Congenital	Temporopolar disconnection	NA	Free/5 y
	36, M	L temporal	Congenital	Temporopolar disconnection	NA	Free/15 Mo
Saavalainen et al., 2015 [15]	22, M	L temporal	NA	ATL + disconnection	Gliosis	Free/2.5 y
	43, F	L temporal	NA	ATL + disconnection	Gliosis	Free/2.1 y
	45, F	R temporal	NA	ATL + disconnection	Gliosis	Engl II/0.79 y
	45, F	L temporal	NA	ATL + disconnection	Gliosis	Engle II/1.2 y
	32, M	L temporal	NA	ATL + disconnection	Gliosis	Engle 3 A/1.3 y
	40, M	R temporal	NA	ATL + disconnection	Gliosis	Free/1.1 y
	30, M	L temporal	NA	ATL + disconnection	Gliosis	Free/3 Mo
Saavalainen et al., 2015 [15] *	33, M	R temporal	NA	ATL + AH	Gliosis	Aura only/4.9 y
	31, M	R temporal	NA	ATL + AH	Gliosis	Free/6.2 y
	43, F	R temporal	NA	ATL + AH	Gliosis	Free/5.9 y
	44, F	R temporal	NA	ATL + AH	Gliosis	Free/3.2 y
	43, M	L temporal	NA	ATL + AH	Gliosis	Free/3.8 y
Panov et al., 2016 [14]	24, F	L temporal	NA	Disconnection (concern for verbal memory)	Mild gliosis	Free
	50, M	R temporal	NA	Lesionectomy (patient preference)	Reactive astrogliosis	Free
	45, F	R temporal	NA	ATL + AH (Szs originated from HC)	Severe astrogliosis	Free
	23, M	R temporal	NA	ATL +AH (increased volume or amygdala)	Mild gliosis	Engle IIb (recurred Szs but overall improved)
	45, F	R temporal	NA	ATL +AH (increase volume and T2 MRI signal of HC)	Mild Chaslin’s gliosis, with FCD-IC affecting HC	Free
	39, M	R temporal	NA	ATL +AH (No reported MRI or EEG early involvement of HC)	Moderate gliosis and mild astrogliosis of HC	Free
De Souza et al., 2018 [16]	18, F	L temporal	NA	ATL + amygdalectomy alone	Architectural disorganization suggestive of FCD	Free/1 y

AH: amygdalohippocampectomy, ATL: anterior temporal lobectomy, EEG; electroencephalography, F: female, FCD-IC: focal cortical dysplasia, with abnormal radial and tangential cortical amination according to Blumcke’s classification, HC: hippocampus, L: left, M: male, Mo: month, MRI: magnetic resonance imaging, NA: not available, R: right, Szs: seizures, T2: Time 2, y: year. * Samples from the base of the encephalocele were obtained from all 12 surgically treated patients. All of these samples showed gliosis and five patients (42%) had mild cortical laminar disorganization. The temporal lobe samples showed gliosis in 11 patients and heterotopic neurons in four patients. Hippocampus and amygdala were resected in five patients; the samples revealed normal mesial structures (*n* = 2), gliosis (*n* = 2), or small focal neuronal loss in CA2 area interpreted as possible mild hippocampal degeneration (*n* = 1).

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
