# Peer review of "Epilepsy Surgery for Skull-Base Temporal Lobe Encephaloceles: Should We Spare the Hippocampus from Resection?"

_brainsci, 2018, doi:10.3390/brainsci8030042_

Reviewer 1 Report

 This is a very interesting case report dealing with the possibility and benefit of hippocampus sparing during epilepsy surgery for skull-base encephaloceles.

Method description and outcome description and discussion are sound. I believe however that introduction should be reinforced by some more recent references.

English and data presentation are adequate and need no further improvement.

Author Response

Dear Reviewer:

We greatly appreciate your valuable input to our paper. Many thanks for all the suggestions you have made in order to improve our publication. Here is my response to your suggestions:

1)      Regarding reinforcing the introduction with newer references, I have reviewed the midecal literature, as suggested by one of the reviewers, and provided a table surgically treated temporal lobe encephaloceles in patients with epilepsy. I have included two most recent references number 14 and 13 from 2015 and 2016 (line 64) respectively and another reference (line 72) number 15 from 2018.

Many thanks for your valuable input.

Please let me know if you have any further suggestions or comments.

Once again, thank you.

Firas

Reviewer 2 Report

Overall, I think the paper by Bannout et al. is a nice description of epilepsy surgery for a rare condition--temporal encephaloceles. This is a very small "series" of 2 patients and doesn't add much to our understanding of this disease. Yet, because these are rare cases, the publication is nevertheless warranted.

One thing that would make this paper far more useful to readers is a table showing all the prior temporal lobe encephalocele papers and their characteristics (e.g., number of patients, outcomes, etc.). That would be a beautiful way to succinctly describe the literature this paper hopes to bolster with its additional two patents. 

Concerning the discussion, the Bannout et al. state how their results show that lesionectomies might be more appropriate for these patients than temporal lobectomies. They cite the UCSF study by Panov et al., where 4 of 6 patients were treated with lobectomies as their foil. But this is a false dichotomy. The Panov et al. study did two lesionectomies, exactly as many as Bannout et al. And the reason they did lobectomies in the other four cases was because ECoG showed wide temporal or hippocampal involvement. Bannout et al. also use ECoG to tailor their resection, but were lucky that the two cases they showed only had the peri-lesional area involved. The upshot is that Bannout et al.'s argument about doing a tailored resection is something other people are already doing. They should acknowledge this in the Discussion.

Lastly, please crop the EEG image in Fig. 6 so it no longer includes the Windows Start Bar on the bottom.

Author Response

Dear Reviewer:

We greatly apperacitae your valuable input to our paper. Many thanks for all the suggestions you have provided us with in order to improve our publication. Here is my response to your following suggestions:

1)      Regarding reviewing the medical literature, we have reviewed the medical literature for surgically treated temporal lobe encephaloceles in patients with epilepsy and added a table with all reported cases in the English literature. I have exluded one case in Korean and another one in Russian. Subsequently I had to limit the manuscript title to "temporal encephalocele".

2)      We have acknowledged in our first manuscript discussion that tailored resection has been described in the medical literature by Giulioni (line 183). Please let me know if this is not enough. This is a very good point. 

3)      I have modified using the valuable reference of Panov to address your point that ATLAH can be considered when there is an electrographic or radiographic evidence of wide mesial structural involvement (line 197). 

Once again, many thanks for your valuable input.

Firas